# Nasopharyngeal Carcinoma Radiomic Evaluation with Serial PET/CT: Exploring Features Predictive of Survival in Patients with Long-Term Follow-Up

**DOI:** 10.3390/cancers14133105

**Published:** 2022-06-24

**Authors:** Adam A. Dmytriw, Claudia Ortega, Reut Anconina, Ur Metser, Zhihui A. Liu, Zijin Liu, Xuan Li, Thiparom Sananmuang, Eugene Yu, Sayali Joshi, John Waldron, Shao Hui Huang, Scott Bratman, Andrew Hope, Patrick Veit-Haibach

**Affiliations:** 1Department of Medical Imaging, Sunnybrook Health Sciences Centre, University of Toronto, Toronto, ON M4N 3M5, Canada; adam.dmytriw@gmail.com (A.A.D.); reut.anconina@sunnybrook.ca (R.A.); 2Joint Department of Medical Imaging, Toronto General Hospital, University Health Network, University of Toronto, Toronto, ON M5G 2C4, Canada; claudia.ortega@uhn.on.ca (C.O.); ur.metser@uhn.ca (U.M.); poriferax@hotmail.com (T.S.); eugene.yu@uhn.ca (E.Y.); sayjo87@gmail.com (S.J.); 3Department of Biostatistics, Princess Margaret Cancer Centre, University Health Network, University of Toronto, Toronto, ON M5G 2C1, Canada; zhihuiamy.liu@uhnresearch.ca (Z.A.L.); zijin.liu@uhnresearch.ca (Z.L.); xuan.li@uhnresearch.ca (X.L.); 4Department of Diagnostic and Therapeutic Radiology, Faculty of Medicine Ramathibodi Hospital, Mahidol University,270 Rama VI Road, Ratchathewi, Bangkok 10400, Thailand; 5Department of Radiation Oncology, Princess Margaret Cancer Centre, University Health Network, University of Toronto, Toronto, ON M5G 2C1, Canada; john.waldron@rmp.uhn.ca (J.W.); shaohui.huang@rmp.uhn.ca (S.H.H.); scott.bratman@rmp.uhn.ca (S.B.); andrew.hope@rmp.uhn.ca (A.H.)

**Keywords:** nasopharyngeal carcinoma, radiomics, FDG-PET/CT, otolaryngology, radiation oncology

## Abstract

**Simple Summary:**

Nasopharyngeal carcinoma (NPC) is a frequent head and neck cancer, especially in Asian countries. Our studies investigated the value of minable data derived from standard of care PET/CT imaging in patients with NPC. The here presented evaluation found that certain specific imaging features in this patient population can be potentially used to predict overall survival and progression free survival at different time points in those patients.

**Abstract:**

Purpose: We aim determine the value of PET and CT radiomic parameters on survival with serial follow-up PET/CT in patients with nasopharyngeal carcinoma (NPC) for which curative intent therapy is undertaken. Methods: Patients with NPC and available pre-treatment as well as follow up PET/CT were included from 2005 to 2006 and were followed to 2021. Baseline demographic, radiological and outcome data were collected. Univariable Cox proportional hazard models were used to evaluate features from baseline and follow-up time points, and landmark analyses were performed for each time point. Results: Sixty patients were enrolled, and two-hundred and seventy-eight (278) PET/CT were at baseline and during follow-up. Thirty-eight percent (38%) were female, and sixty-two patients were male. All patients underwent curative radiation or chemoradiation therapy. The median follow-up was 11.72 years (1.26–14.86). Five-year and ten-year overall survivals (OSs) were 80.0% and 66.2%, and progression-free survival (PFS) was 90.0% and 74.4%. Time-dependent modelling suggested that, among others, PET gray-level zone length matrix (GLZLM) gray-level non-uniformity (GLNU) (HR 2.74 95% CI 1.06, 7.05) was significantly associated with OS. Landmark analyses suggested that CT parameters were most predictive at 15 month, whereas PET parameters were most predictive at time points 3, 6, 9 and 15 month. Conclusions: This study with long-term follow up data on NPC suggests that mainly PET-derived radiomic features are predictive for OS but not PFS in a time-dependent evaluation. Furthermore, CT radiomic measures may predict OS and PFS best at initial and long-term follow-up time points and PET measures may be more predictive in the interval. These modalities are commonly used in NPC surveillance, and prospective validation should be considered.

## 1. Introduction

Molecular imaging with positron emission tomography (PET)/computed tomography (CT) using radiopharmaceutical ^18^F-fluoro-deoxy-glucose (FDG) has a prominent role in the detection and staging of nasopharyngeal carcinoma (NPC) [1,2,3,4,5,6,7]. A widely used parameter for lesion characterization is the standardized uptake value (SUV) measured in FDG PET/CT scans. The mean assessment of tumor activity provided by this measurement has been shown to predict tumor aggressiveness and response to treatment in a variety of cancers [8,9]. The underlying spatial distribution of tracer activity within a tumor is, however, not well described by this parameter. Indeed, NPC can be heterogeneous owing to locoregional differences in cell density and blood supply, as well as cell growth and cell death. This ‘intra-tumor heterogeneity’ can be measured by extraction of radiomics features by analyzing the variation in the spatial arrangements, thereby representing an unequal distribution of tracer activity within a tumor [10,11,12]. It has been described that tumor heterogeneity correlates with tumor aggressiveness and disease outcome in a variety of tumor types [13,14,15,16]. Hence, a radiomic biomarker can be generated from molecular (as well as anatomical) images and possibly provide improved prognostic information regarding tumor behavior and/or disease outcome.

Radiomics refers to the high-throughput extraction of quantitative imaging features from medical images, including measures of tumor surface irregularity and texture heterogeneity, with the intent of creating mineable databases to build descriptive and predictive models relating image features to phenotypes or gene–protein signatures [17]. With increasing advances in the last years, the translational potential of this process in oncology has been demonstrated [18], including areas such as evaluation of prognosis [19,20,21,22] chance of survival [23,24,25], recurrence [26], distant metastasis [27,28], post-radiotherapy pneumonitis [29], staging [30] and screening [31]. The essence of radiomics is the extraction of high-dimension feature data to quantitatively describe attributes of volumes of interest. These can be classified in many ways, including the division between semantic features, which are those commonly used by radiologists to describe regions of interest; its shape and size; and agnostic features, which are those obtained through quantitative descriptors. The latter may be further divided into first-, second- and higher-order statistical outputs [32].

While local tumor control with chemoradiotherapy is excellent, distant failure occurs in up to 20% of these patients; however, it often occurs with considerable latency [33]. The inclusion of radiomic data in the assessment of NPC has not been yet validated. There are only very few studies that investigated the combined prognostic value of PET-based radiomics combined with CT-based radiomics [13,14]. This is, however, an important aspect because both imaging modalities are acquired at the same time within combined PET/CT. Initial studies have shown promising results in terms of nomogram building, risk stratification and some have identified possible prognostic radiomics parameters [34,35]. However, there are even fewer studies available looking at the long-term prognostic value of combined PET and CT radiomics [36]. 

The goal of this study, therefore, was to develop an FDG PET/CT radiomic biomarker based on a retrospective analysis of serially acquired FDG PET/CT scans in patients with known diagnosis of NPC in patients with long-term follow up data. The intra-tumor image-based heterogeneity has been shown to correlate with clinical outcomes such as treatment response and survival in a variety of tumor types, including several ENT-cancer types [13,14]. This suggests that a radiomic biomarker that quantifies the spatial distribution of tracer activity within a tumor on FDG PET/CT can be developed and cross-referenced with follow-up to better predict outcomes and influence evidence-based clinical decision making in patients with NPC. 

We hypothesize that prognostic information derived by pre-therapy and serial follow up hybrid FDG PET/CT image-based data via radiomics can improve long-term outcome prediction and prognostic stratification in patients with NPC. Such stratification may allow clinicians to better tailor their treatment protocols to each patient’s risk profile.

## 2. Materials and Methods

### 2.1. Study Cohort

This study was conducted retrospectively. Approval by the institutional research ethics board was obtained as indicated below. REB waived the requirement for informed consent.

All newly diagnosed for nasopharyngeal carcinoma patients treated with curative intent at our institution from 2005 to 2006 were reviewed. Patients with available serial FDG PET/CT (PET/CT) for staging of NPC were included. Serial PET-imaging was performed at baseline and every three months up until 15 month (timepoints 1–5). 

Patients were identified from an in-house prospective Head and Neck Bioclinical Anthology of Outcomes System where baseline demographic, TNM staging, treatment and outcome data were collected at point-of-care using Formatted Anthology Synoptic Tick sheet process [37]. Vital status data were further supplemented by linkage to provincial cancer registry.

### 2.2. Image Acquisition

FDG PET/CT imaging was obtained according to our standard of care institutional imaging protocol at the time [38]. Patients were asked to avoid exercise for 24 h and fast for 6 h before the examination. Patients received an IV injection of 5 MBq/kg (range of 250–550 MBq) of FDG. Oral contrast was applied for bowel opacification in all patients. No intravenous contrast was applied in this patient population. Dedicated PET image acquisition for head and neck was performed using 2 6-minute beds with both arms down. PET images were reconstructed iteratively with scatter correction (6 iterations and 16 subsets) with Fourier rebinning (FORE), using a 5 mm full width at half maximum (FWHM) gaussian filter and a 128 matrix. All PET/CT procedures were acquired on the same system (Biograph). The CT was acquired with an effective 100 mAs and 130 kV (CareDose), slice width/collimator of 5.0 mm/2.5 mm, a rotation time of 1 s and a feed/rotation of 5 mm, with a recontruction increment 3 mm and a slice thickness of 3 mm. 

### 2.3. Radiomics and Statistical Analysis

We quantified intra-tumor heterogeneity retrospectively by performing radiomics feature analysis from previously acquired FDG PET/CT images using LIFE X version 6.1 software (lifexsoft.org) [39] via the quantitation of various radiomics features based on the spatial arrangement and variation of pixel intensities within a defined volume of interest. The standard setup of the LIFE X software was used without additional manipulation or interpolation. Absolute resampling with defined boundaries was used (e.g., for PET, the min value was 0 and the max was 20). The number of grey levels was 64. Thus, the binning method was fixed among the scans. Concerning the size of bins, the following formula is used by the program: Number of Grey Levels = ((Max boundary−Min Boundary)/Bin Size) + 1. The fixed calculated values for the bin size were 0.3125 for PET and 10 for CT. 

PET volumes of interest (VOI) were defined based on (a) background threshold, (b) threshold at 40% and (c) threshold at 70% of the SUVmax [40]. All radiomics features were evaluated for all threshold groups. Since there is no thresholding method available for the CT-component, the contours for the CT-derived VOI were performed manually in slice-by-slice fashion to cover the entire tumor. No automated propagation method was used for CT contouring. The minimal VOI included at least 64 voxels and was confirmed (by the “CheckTex” feature in the software) to make sure it created a single contiguous VOI that enabled consistent textural feature calculation. No PET-positive findings outside of the actual tumors were included into the VOI. The PET-volume delineation was performed by a dual certified radiologist/nuclear medicine physician with >15 years clinical experience. The CT-component was delineated by a board-certified radiologist with >7 years of clinical imaging experience. 

Radiomics features analysis included the following: conventional metrics features reporting the mean, median, maximum, minimum values of the voxel intensities on the image, size and shape histogram-based features such as volume, compacity and sphericity including their asymmetry (skewness), flatness (kurtosis), uniformity and randomness; and textural features (such as GLCM (Gray-Level Co-occurrence Matrix), GLRLM (Grey-Level Run Length Matrix), NGLDM (Neighborhood Grey-Level Different Matrix) and GLZLM (Grey-Level Zone Length Matrix)). 

The data cutoff date for outcome analysis was April 29 2021. Summary statistics were used to describe patient, disease and treatment characteristics. The clinical endpoints of interest were progression free survival (PFS) and overall survival (OS). The actuarial rates of OS and PFS were estimated using the Kaplan–Meier method. Univariable Cox proportional hazards models were used to assess prognostic factors, including clinical variables and PET/CT radiomic features, for death and relapse. *p*-values ≤ 0.05 and ≤0.01 were considered statistically significant for clinical and radiomic features, respectively, to account for the larger number of features evaluated for the latter. First, to account for a larger number of radiomic features assessed compared to the clinical variables, we set the statistical significance level for radiomic feature selection at a more stringent *p* ≤ 0.01. Secondly, among those radiomic features with a *p*-value < 0.01, we filtered out highly correlated features by assessing their pairwise correlation coefficients, using the algorithm described below. Due to the limited number of deaths and progression events, multivariable analyses were not attempted. All analyses were carried out in R (version 4.0.2).

The pre-processing of the radiomic features included standardization and removing features with more than 40% missing values. Two models were developed and compared: a ‘baseline model’ and a ‘time-dependent model’. In the baseline model, univariable Cox proportional hazards models were fitted to evaluate association of each baseline radiomic feature with OS and PFS. In contrast, a time-dependent Cox proportional hazards model took into account radiomic features from all time points as time-varying covariates such that the feature values were updated at each time point. Moreover, multiple comparisons were considered in our study; in both statistical approaches (time-dependent Cox model and landmark analysis), we used a more stringent threshold for statistical significance. In the former approach, we also filtered out highly correlated features to arrive at the reduced final set. The Bonferroni correction to control for familywise error rate was not used, because it would be too conservative given that the features are highly correlated.

Among the significant features (*p*-value ≤ 0.01) based on the univariable models, we used the following algorithm to determine those that are not highly correlated. The feature with the smallest *p*-value was taken into the “selected set” and the remaining features were considered as candidate features (Step 1). We then calculated Pearson’s correlation coefficient ρ between the selected feature and each candidate feature, and those with a ρ < 0.7 were considered as new candidate features (Step 2). We chose one feature with the smallest *p*-value in the candidate features of Step 2 into the “selected set” (Step 3). Steps 2 and 3 were repeated until there were no more candidate features. Finally, we presented the final “selected set”.

Lastly, a landmark analysis was conducted. We considered each time point when the PET/CT scans took place as a landmark, or new baseline, and univariable Cox proportional models were used to evaluate radiomic features for each landmark, with the ‘risk set’ (i.e., patients who had not experienced the outcome) updated at each time point. A *p*-value ≤ 0.01 was considered statistically significant. Leave-one-out cross-validated C-index was calculated.

## 3. Results

### 3.1. Baseline Characteristics of Patients and Tumors

A total of 60 patients with nasopharyngeal carcinoma (23 female, 37 male) with serial imaging were included. Overall, 278 PET/CT were acquired in these patients and 52 patients had five serial PET/CTs. All patients underwent a baseline PET/CT for staging. Eighteen patients died during the follow-up time, and eleven of those patients died from direct sequalae of their index cancer. The median age for all patients was 51.2 (range 18.3–74.8 years). The majority were non-smokers (71%, 41 patients). Tumor volumes ranged from 4.3 cm^3^ to 140 cm^3^ (average: 37.3 cm^3^). All patients were evaluated clinically to have a curative therapeutic intent. Baseline characteristics and treatment information are shown in Table 1. The majority of patients had stage III disease (33%, 20 patients).

### 3.2. Treatment Outcomes

Of 60 patients, 4 experienced local failures, 5 experienced regional failures, and 8 experienced distant failures over the study period. Eighteen patients were deceased at the end of 2020. Median follow-up was 11.72 years (range 1.26–14.86). The estimated 5-year and 10-year PFSs were 80.0% (95% CI: 70.5–90.8%) and 66.2% (95% CI 55.1–79.4%), and OS was 90.0% (95% CI: 82.7–97.9%) and 74.4% (95% CI 64.0–86.5%), as shown in Figure 1.

### 3.3. Time-Dependent vs. Baseline Cox Model

On univariable analysis, baseline radiomic analysis was not significantly associated with OS for any parameter. Time-dependent modeling showed that CT_NGLDM_Busyness (HR 2.54, 95% CI 1.29–5), PET_CONVENTIONAL_SUVbwmax (HR 2.66, 95% CI 1.56–4.55) and PET_GLZLM_GLNU (HR 2.26, 95% CI 1.46–3.49) for a 40% threshold as well as CT SHAPE_Volume.vx (HR 1.94. 95% CI 1.34–2.8) and PET_DISCRETIZED_SUVbwmax (HR 2.74, 95% CI 1.58–4.74) for a 70% threshold were significantly associated with OS (Table 2). 

Similarly, baseline radiomic analysis was not significantly associated with PFS for any parameter. Again, time-dependent modeling showed that PET_DISCRETIZED_SUVbwpeakSphere0.5mL (HR 2.06, 95% CI 1.28–3.31) and PET_GLZLM_GLNU (HR 1.67 95% CI 1.23–2.26) for a 40% threshold as well as PET_CONVENTIONAL_SUVbwQ1 (HR 1.84, 95% CI 1.23–2.76) and PET_CONVENTIONAL_TLG.mL (HR 5.67, 95% CI 1.75–18.39) were significantly associated with PFS. The full list of significant features are shown in Appendix A. 

### 3.4. Landmark Analyses

On landmark modeling, CT radiomic parameters were significantly associated with OS for time point 5, PETs at 40% threshold parameters were significantly associated with OS for time points 2, 3 and 5 and PETs at 70% threshold parameters were significantly associated with OS for time point 3. 

CT radiomic parameters were significantly associated with PFS for time point 5, PET at 40% threshold parameters were significantly associated with PFS for time point 2 and PET at 70% threshold parameters were significantly associated with PFS for time points 2 and 4. These are elaborated in Table 3 and Table 4 as well as in Figure 2 and Figure 3.

## 4. Discussion

Our study evaluated radiomics features from serial, combined PET and CT and evaluated the prognostic value of specific radiomic parameters for OS and PFS. Furthermore, we evaluated a baseline evaluation model vs. a time-dependent model. Finally, we showed, in a landmark evaluation, the value of individual radiomics parameters at different time points in correlation with long-term follow up. 

A major strength of radiomics is that it can be applied to standard-of-care radiological exams, providing a potentially massive amount of information for the construction of databases and clinical-decision support systems. However, as with other quantitative image analyses, variations in the acquisition and reconstruction parameters can introduce changes that are not due to underlying biological effects [32]. For this reason, multiple efforts to standardize these parameters have been attempted [41,42], including the National Cancer Institute’s Quantitative Imaging Network (QIN) [43] and the Radiological Society of North America’s/National Institute for Biomedical Imaging and Bioengineering’s Quantitative Imaging Biomarkers Alliance (QIBA) [44].

The other challenges of the radiomics are in the definition of the volumes of interest to be analyzed [45,46], which may include the primary tumor, satellite lesions, nodal and distant metastases; and in their segmentation, which is especially challenging, given the sometimes indistinct borders of many tumors (and their subregions), the difficulty to define the ground truth for these analyses and the varying degrees of automation versus human input to be used in the process [32]. It is well recognized that interoperator variability of manually contoured tumors is high [47,48], but this input is currently most often necessary in cancer cases, due to their inter- and intrasubject morphological and functional heterogeneity [32].

The resulting features should be associated with other clinical, genomic and histopathological data, including the desired end points to be evaluated and potential causal or confounding factors, in order to produce mathematical models of outcomes. This can be achieved through supervised data analysis, which include statistical and machine learning methods, such as neural networks, linear regression and Cox proportional hazards regression [17].

### 4.1. Nasopharyngeal Carcinoma

The standard workup of patients with NPC often includes contrast-enhanced computed tomography (CT) and/or magnetic resonance imaging (MRI) [49]. Additionally, ^18^F-fluoro-2-deoxy-D-glucose (FDG) positron emission tomography (PET) imaging has been used for several years in this disease for staging as well [1,2,3,4,5,6,7]. Most radiomics studies on these tumors have focused on CT and MRI images, including the development of radiomics signatures that outperforms TNM staging when evaluating overall survival [19,50,51,52], with external validation in a separate study [21]; they are predictors of the tumor’s HPV status [53,54]. However, studies evaluating PET radiomics have demonstrated a potential to predict the risk of local failure [55,56,57,58] and distant metastasis [57,58], progression-free [59,60] and overall survival [57,58,59,60].

Nasopharyngeal epithelial carcinoma has some particular characteristics when compared to other head and neck tumors, including its association with the Epstein–Barr virus, with mostly Southeast Asian prevalence [61] and high radiosensitivity [62] and most often not needing surgical resection. Its workup also often includes CT, MRI and FDG-PET/CT and certain features from the latter modality, including SUVmax, metabolic tumor volume (MTV) and total lesion glycolysis (TLG), were associated with event-free survival and overall survival in recent meta-analyses [8,9]. 

The radiomics studies evaluating this tumor entity are significantly more limited, especially when considering the mature follow-up data that we used for endpoint correlation.

Many studies are using pretreatment MRI features, which have been used to predict progression-free survival in stage III-IVb patients [63,64,65] and in a smaller sample including low grade tumors [66]. A few earlier studies tried to measure FDG uptake heterogeneity by using different quantitative measures, including SUVmax/SUVmean [67], SUV standard deviation/SUVmean [68] and the derivative of the volume-threshold function [69]; however, a more recent study [13] using the radiomics approach found skewness (a first-order feature) to be a predictor of relapse-free survival and uniformity (a texture feature) to be a predictor of overall survival, which improved the prognostic stratification of other risk factors found by using multivariate analysis, namely age and serum EBV DNA load. A large multicenter study recently showed that treatment decision and prognosis for specific stages could reliably based on a radiomic based nomogram [70]. However, this study again used MRI for imaging. One study using PET/CT evaluated the value of radiomics for prediction of distant metastases and local recurrence [14]. The difference to our study was, however, that only higher stage tumors were evaluated and only PET-features and uncombined PET and CT parameters were evaluated. A recent PET/MR study evaluated the combined value of PET and MR parameters for staging (however, not for prognosis) and also found that radiomic parameters from both modalities (PET and MR) are useful for the evaluation of NPC [71].

### 4.2. Overall and Progression-Free Survival

On landmark analysis, histologic kurtosis was a significant predictive factor for worse OS. Furthermore, histologic kurtosis on PET is the most predictive for stage IV disease and may be valuable for prognostication in patients with advanced-stage NPC. On CT, the HR for LZHGE was the radiomic parameter, which was significant. By contrast, this was most predictive for stage I/II/III disease and may be valuable for prognostication in patients with early stages of NPC.

Time-dependent modeling over the robust 15-year follow-up of this cohort further suggested that only PET gray-level zone length matrix (GLZLM) gray-level non-uniformity (GLNU) is highly significantly associated with OS. The use of GLZLM and GLNU has been shown to be a reliable differentiator between nasopharyngeal as well as hepatic carcinoma and lymphoma, whereas this is the first study to suggest an association with OS itself. The fact that these PET parameters appear to be best applied to late-stage disease may explain why this radiomic measure relates better to OS than PFS.

To the best of our knowledge, this is the only radiomics study in patients with NPC showing the value of landmark analyses. On this evaluation, CT radiomic parameters were significantly associated with both OS and PFS for time points 1 and 5 whereas PET at 40% and PET at 70% threshold parameters were significantly associated with OS and PFS principally across time points 2, 3 and 4. Many of these associations were very highly associated, and this novel finding is of uncertain significance. It could be speculated that CT radiomics parameters, such as kurtosis, neighborhood grey-level different matrix (NGLDM), grey level co-occurrence matrix (GLCM) and gray level run length matrix (GLRLM), could be of optimal prognostic use at first diagnosis and late follow-up. By contrast, PET radiomics parameters, such as GLRLM and NGLDM but also SUV measures, could be of optimal prognostic use at interval follow-up points between these two termini. This could relate to the molecular/metabolic activity related both to progression and response to treatment.

Our study has several limitations. The study was conducted retrospectively and in a relatively small cohort of overall only 60 heterogeneous patients. We did not split our cohort into training and validation data sets since the cohort would have been too small for any meaningful radiomics evaluation. Moreover, this is an evaluation from a single institute; thus, the findings may be restricted to our applications rather than a more general clinical setting.

Further studies are needed to validate our findings. Since PET/CT acquisition occurred a long time ago, current PET/CT imaging is mostly performed with more modern reconstructions. Another partial limitation to our study is that the patient cohort is mostly reflective of an endemic patient population and, therefore, may be generalizable to WHO type I patients. 

Moreover, there are always several technical considerations that can be discussed in radiomics research. For example, since some of the radiomics features can be voxel size and gray-level discretization-dependent (at least in CT), voxel-size resampling has been found as an appropriate pre-processing step to obtain more reproducible CT features [72]. Moreover, this is a certain variation in PET textural features relative normal stochastic image variations, and it has been described in the literature, which can be additionally feature-dependent [73]. 

However, we only used one PET/CT scanner without any variations in acquisition or reconstruction between the evaluated scans. Moreover, we believe that any required pre-processing potentially hampers implementation in clinical routines, unless it is automatically set up in a wide variety of different types of scanners. We chose manual delineation methods for contouring in our study. While there is evidence in the literature that the radiomics evaluation can be influenced by the delineation method (manual vs. automated), we did not have such a program available for CT. For the PET-evaluation, a semiautomated delineation method was used. We chose to use overall survival as an outcome of interest because it is the most reliable and available survival measure. In addition, statistical power would be lower using cancer-specific deaths compared to deaths from any cause due to the reduced number of outcome events.

In any case, we present unique data with different statistical evaluations and a very mature follow-up period. Finally, it is worthwhile to point out that while there are certainly several publications on NPC radiomics available in the literature, this is one of the very few evaluating PET- and CT-combined radiomics. Furthermore, even fewer manuscripts correlate radiomics with such a mature follow up.

## 5. Conclusions

This study with very mature follow-up data on NPC suggests that mainly PET-derived radiomics features are predictive for OS but not PFS in a time-dependent evaluation. Furthermore, CT radiomic measures may predict OS and PFS best at initial and long-term follow-up time points, and PET measures may be more predictive in the interval. These modalities are already commonly used in NPC surveillance and prospective validation should be considered.

## Figures and Tables

**Figure 1 cancers-14-03105-f001:**
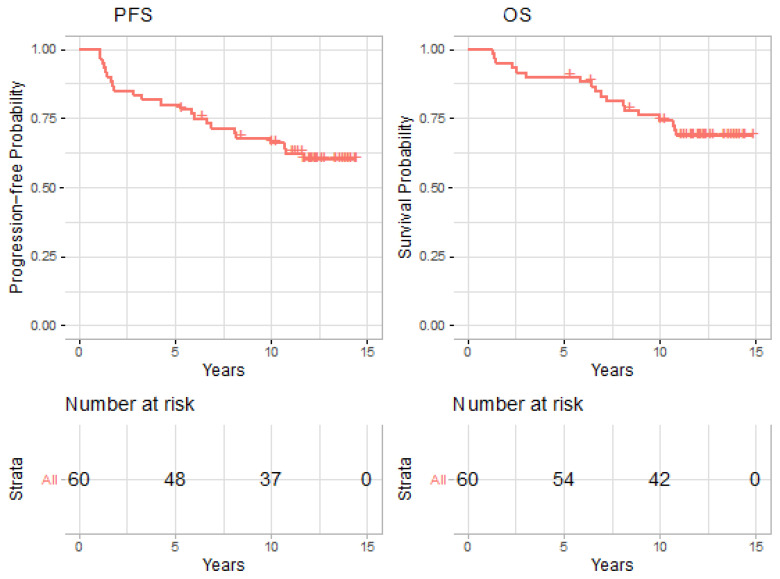
Kaplan-Meier plots for progression free survival (PFS) and overall survival (OS).

**Figure 2 cancers-14-03105-f002:**
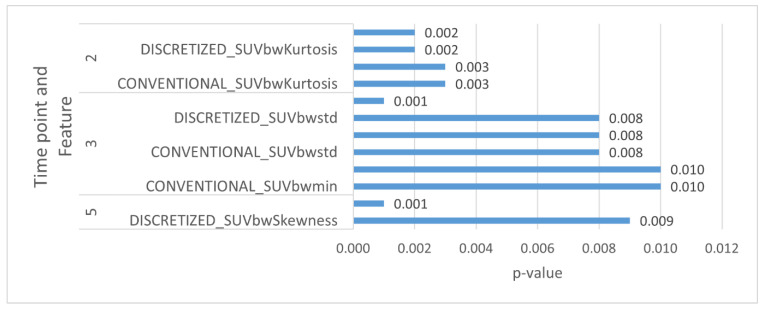
Selected PET and CT radiomic parameters at different time points for OS.

**Figure 3 cancers-14-03105-f003:**
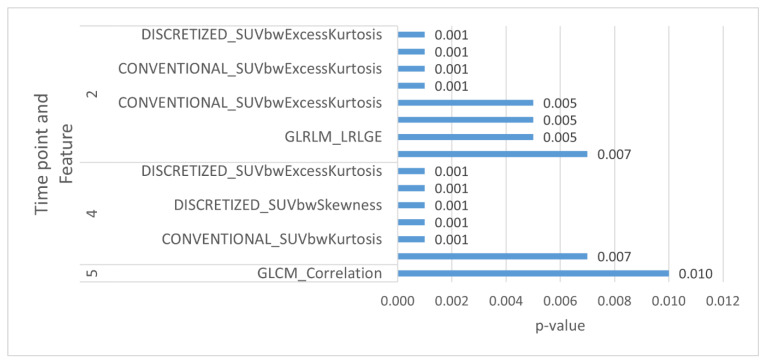
Selected PET and CT radiomic parameters at different time points for PFS.

**Table 1 cancers-14-03105-t001:** Clinicopathologic characteristics.

Characteristic	*n* = 60 (%)
Age, median (range), years	51.2 (18.3–74.8)
Male Female	37 (62%) 23 (38%)
Smoking Status	
Current	4 (8%)
Former	13 (22%)
Non-smoker	41 (71%)
Unknown	2 (4%)
Pathology	
WHO I	1 (2%)
WHO IIA	10 (16%)
WHO IIB	49 (82%)
Stage	
I	2 (4%)
II	11 (21%)
III	19 (37%)
IVa	12 (23%)
IVb	7 (13%)
IVc	1 (2%)
T	
T1	24 (40%)
T2	12 (20%)
T3	7 (12%)
T4	17 (28%)
N	
N0	4 (7%)
N1	22 (20%)
N2	26 (43%)
N3	1 (2%)
N3A	5 (8%)
N3B	2 (3%)
M	
M0	57 (95%)
M1	3 (5%)
Status	
Alive	42 (70%)
Dead	18 (30%)
Treatment	
Radiotherapy alone	6 (10%)
Chemoradiotherapy	54 (90%)
Local Failure	4 (7%)
Regional Failure	5 (8%)
Distant Failure	8 (13%)

**Table 2 cancers-14-03105-t002:** Comparison of time-dependent model with selected features vs the baseline model.

Overall Survival		Time-Dependent Model		Baseline Model	
Modality	Radiomic Features	HR (95%CI)	*p*-Value	HR (95%CI)	*p*-Value
CT + PET 40%	CT_NGLDM_Busyness	2.54 (1.29, 5.00)	0.0069	0.98 (0.73, 1.33)	0.90
	PET_CONVENTIONAL_SUVbwmax	2.66 (1.56, 4.55)	0.0004	1.08 (0.74, 1.57)	0.69
	PET_GLZLM_GLNU	2.26 (1.46, 3.49)	0.0002	1.14 (0.95, 1.37)	0.16
CT + PET 70%	CT_SHAPE_ Volume.vx.	1.94 (1.34, 2.80)	0.0004	1.08 (0.85, 1.38)	0.51
	PET_DISCRETIZED_SUVbwmax	2.74 (1.58, 4.74)	0.0003	1.07 (0.70, 1.64)	0.74
**Progression-Free Survival**		**Time-Dependent model**		**Baseline Model**	
**Modality**	**Radiomic Features**	**HR (95%CI)**	***p*-Value**	**HR (95%CI)**	***p*-Value**
CT + PET 40%	PET_DISCRETIZED_ SUVbwpeakSphere0.5mL	2.06 (1.28, 3.31)	0.0029	1.08 (0.74, 1.58)	0.68
	PET_GLZLM_GLNU	1.67 (1.23, 2.26)	0.0011	1.09 (0.90, 1.31)	0.38
CT + PET 70%	PET_CONVENTIONAL_SUVbwQ1	1.84 (1.23, 2.76)	0.0031	1.05 (0.76, 1.43)	0.78
	PET_CONVENTIONAL_TLG.mL	5.67 (1.75, 18.39)	0.0039	1.14 (0.68, 1.91)	0.62

**Table 3 cancers-14-03105-t003:** Landmark analysis for overall survival.

Modality	Time Point	Significant Features	HR (95% CI)	*p*-Value	C-Index
CT	5	GLCM_Correlation	0.33 (0.17, 0.62)	0.001	0.792
PET 40%	2	CONVENTIONAL_SUVbwKurtosis	1.85 (1.23, 2.78)	0.003	0.616
	2	CONVENTIONAL_SUVbwExcessKurtosis	1.85 (1.23, 2.78)	0.003	0.616
	2	DISCRETIZED_SUVbwKurtosis	1.94 (1.27, 2.97)	0.002	0.653
	2	DISCRETIZED_SUVbwExcessKurtosis	1.94 (1.27, 2.97)	0.002	0.653
	3	CONVENTIONAL_SUVbwstd	2.01 (1.2, 3.38)	0.008	0.551
	3	DISCRETIZED_SUVbwstd	2.01 (1.2, 3.38)	0.008	0.538
	3	GLZLM_ZLNU	2.3 (1.46, 3.64)	0.001	0.707
	5	DISCRETIZED_SUVbwSkewness	0.32 (0.14, 0.75)	0.009	0.736
PET 70%	3	CONVENTIONAL_SUVbwmin	2.01 (1.19, 3.41)	0.010	0.546
	3	CONVENTIONAL_SUVbwQ3	1.98 (1.17, 3.34)	0.010	0.519
	3	DISCRETIZED_SUVbwstd	1.98 (1.2, 3.27)	0.008	0.536

**Table 4 cancers-14-03105-t004:** Landmark analysis for progression-free survival.

Modality	Time Point	Significant Features	HR (95% CI)	*p*-Value	C-Index
CT	5	GLCM_Correlation	0.46 (0.26, 0.83)	0.010	0.690
PET 40%	2	CONVENTIONAL_SUVbwKurtosis	2.38 (1.54, 3.67)	0.001	0.671
	2	CONVENTIONAL_SUVbwExcessKurtosis	2.38 (1.54, 3.67)	0.001	0.671
	2	DISCRETIZED_SUVbwKurtosis	2.44 (1.57, 3.77)	0.001	0.689
	2	DISCRETIZED_SUVbwExcessKurtosis	2.44 (1.57, 3.77)	0.001	0.689
	2	GLCM_Energy AngularSecondMoment.	1.66 (1.15, 2.41)	0.007	0.648
	2	GLRLM_LRLGE	1.67 (1.17, 2.39)	0.005	0.629

## Data Availability

Study data are available upon reasonable request to the corresponding author.

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
