# Peer review of "Nasopharyngeal Carcinoma Radiomic Evaluation with Serial PET/CT: Exploring Features Predictive of Survival in Patients with Long-Term Follow-Up"

_cancers, 2022, doi:10.3390/cancers14133105_

Round 1
Reviewer 1 Report
The authors detail a number of radiomic features from PETCT, analyzing how these variable correlate with outcome over time in patients with nasopharyngeal carcinoma. While the topic is interesting, I have significant concerns regarding this study and feel that the small sample size significantly limits the findings of this manuscript
1) While typically longer follow up is advantageous, the study would have considerably benefitted from additional patients. It is unclear as to why the study period was restricted from 2005-06. The serial PET imaging requires a minimum of 15 months but the study could have easily included additional patients and still maintained a lengthy follow up period. This is particularly relevant since the majority of progression type events happened within the first 5 years. Per the authors, this limited n prevented any useful multivariate analyses. Given the presence of important covariates that are known to influence outcome, the true association of these radiomic features have with PFS and OS cannot be determined.
2) The utility of these variables would be strengthened if correlation could be shown a recurrence specific variable ( e.g. time to progression) as both PFS and OS can be obscured by non-cancer related death. While non-cancer related death is significantly important when evaluating impact of a certain treatment, in the current study this is of less importance.
3) An additional limitation not mentioned is the study population is largely reflective of an endemic population and may be less generalizable to WHO type I patients.
Author Response
Reviewer 1
The authors detail a number of radiomic features from PETCT, analyzing how these variable correlate with outcome over time in patients with nasopharyngeal carcinoma. While the topic is interesting, I have significant concerns regarding this study and feel that the small sample size significantly limits the findings of this manuscript
1) While typically longer follow up is advantageous, the study would have considerably benefitted from additional patients. It is unclear as to why the study period was restricted from 2005-06. The serial PET imaging requires a minimum of 15 months but the study could have easily included additional patients and still maintained a lengthy follow up period. This is particularly relevant since the majority of progression type events happened within the first 5 years. Per the authors, this limited n prevented any useful multivariate analyses. Given the presence of important covariates that are known to influence outcome, the true association of these radiomic features have with PFS and OS cannot be determined.
The study time frame was restricted to 2005 – 2006 since at this time NPC was considered only in funded registry studies. Thus, the serial PET/CT’s evaluated here were done as part of a trial where the value of PET/CT was evaluated for different purposes. It was only several year later that NPC became publicly funded, and then it was only for staging. Thus, there is no other time point in our jurisdiction where we would have serial PET/CT in NPC patients available in a similarly structured way. The reviewer is certainly right that it would be great to have additional cases to maybe be in a position for multivariate analysis. However, regrettably, we do not have any more cases to be analysed in the same way available. As for the limited number of cases: it is certainly true that this is a significant limitation to our study, which is already acknowledged. This is however also the reason why we chose to evaluate our data with a landmark model so that the reader can get information about potential value of individual radiomic features at different time points.
2) The utility of these variables would be strengthened if correlation could be shown a recurrence specific variable ( e.g. time to progression) as both PFS and OS can be obscured by non-cancer related death. While non-cancer related death is significantly important when evaluating impact of a certain treatment, in the current study this is of less importance.
We thank the reviewer for this comment and acknowledge that all-cause mortality is not as specific as cancer-specific mortality. However we chose to use overall survival as an outcome of interest because it is the most reliable and available survival measure. Consequently our results can be more easily generalized to and validated by other institutions. In addition, statistical power would be lower using cancer-specific deaths compared to deaths from any cause due to the reduced number of outcome events for the earlier. We have now added these considerations in our discussion section. If the reviewer still feels that this number should be provided and the cancer specific mortality should be calculated we would happy to provide that for the readers.
3) An additional limitation not mentioned is the study population is largely reflective of an endemic population and may be less generalizable to WHO type I patients.
We agree partly. As seen in our affiliations, the study was conducted in Toronto. The greater Toronto area currently has 6 million inhabitants, ca. 50% are currently immigrants from all over the world. Thus, we agree that this is not as generalizable, but we do think that we have a somewhat representative mixture of NPC cases in our cohort. A comment has been added to the limitations section.

Reviewer 2 Report
Thank you for giving me the opportunity of reviewing this fascinating retrospective study that addresses a relevant oncological question. It was well conducted and excellently evaluated and described.
I have only one request:
more details on primary therapy should be given (please further subdivide in table 1: chemoradiotherapy: how many patients received neoadjuvant or adjuvant chemotherapy, how many simultaneous radiochemotherapy?). Thank you very much!
Reviewer 3 Report
I would like to congratulate the authors for the study design and development, the topic selected and how it was discussed. The role of radiomic features is a forefront topic for the Oncological arena and further evidence is eagerly awaited.
The authors hypothesized that prognostic information derived by pre-therapy and serial follow-up hybrid FDG PET/CT image-based data via radiomics could improve long term outcome prediction and prognostic stratification in patients with NPC. Thus, targeting to
allow clinicians to better tailor treatment protocols to each patient’s risk profile.
Intra-tumor heterogeneity was quantified retrospectively by performing radiomics feature analysis from previously acquired FDG PET/CT images using LIFE X version 6.1 software (lifexsoft.org), via quantitation of various radiomics features based on the spatial arrangement and variation of pixel intensities within a defined volume of interest.
Radiomics features analysis included: conventional metrics features reporting the mean, median, maximum, minimum values of the voxel intensities on the image, size and shape histogram-based features such as volume, compacity and sphericity including their asymmetry (skewness), flatness (kurtosis), uniformity, and randomness; and textural features (such as GLCM- Gray-Level Co-occurrence Matrix, GLRLM- Grey-Level Run Length Matrix, NGLDM- Neighborhood Grey-Level Different Matrix, GLZLM- Grey-Level Zone Length Matrix). Lastly, a landmark analysis was conducted.
Time-dependent modelling suggested that PET gray-level zone length matrix (GLZLM) gray-level non-uniformity (GLNU) was significantly associated with OS. Landmark analyses suggested CT parameters were most predictive at 15 month, whereas PET parameters were most predictive at time points 3, 6, 9 and 15 month.
The study has several limitations, properly reported by the authors:
- study was conducted retrospectively and in a relatively small cohort. - also, this is an evaluation from a single institute.
On the other hand, one PET/CT scanner was used with no variation in acquisition or reconstruction between the evaluated scans. Delineations approaches were acceptable.
The manuscript is complete in each section, fluent and interesting.
Further evidence is awaited, but this paper should be accepted in its present form.
Round 2
Reviewer 1 Report
Thank you for considering my suggestions. Despite the authors' response, I still have significant concerns about the application of the data.
The lack of disease specific outcomes (time to progression, cancer-specific survival) prevents any reasonable conclusions regarding PET parameters and patient outcome. PET parameters could as easily be associated with patients dying in motor vehicle accidents versus disease related processes based on the available data. Furthermore, the argument that they are underpowered to evaluate any disease specific processes is questionable since there are 4 local failures, 5 regional failures, and 8 distal failures. Even with some overlap within these patients, these numbers are not terribly dissimilar from the number of deaths (18) so one cannot argue that one endpoint is underpowered while another is when the difference between the two is likely single digits. This omission is conspicuous especially when it was specifically requested on previous review.
Even if there were correlations with disease specific entities, without proper multivariate analysis unfavorable PET parameters could merely be a marker of locally advanced disease and this analysis simply recapitulates prognostic information already obtained from TNM staging.
Time-based radiomic analysis of PET imaging is an interesting concept but ultimately I feel this is more appropriate for a imaging specific journal as the correlation to outcome is skeptical.